# Reconstructing Sordida subcomplex (Hemiptera, Reduviidae, Triatominae) phylogeny across species distribution range

**Gabriela Burgueño-Rodríguez[1], Julieta Nattero[2,3,4], Néstor Ríos[1], Romina V Piccinali[2,3], Ana L Carbajal-de-la-Fuente[5,6], Francisco Panzera[1], Catarina Macedo Lopes[7], Patricia A Lobbia[8], Antonieta Rojas de Arias[9], Bruno A Sansoni-Ruidíaz[1], María J Cavallo[10], Claudia S Rodríguez[11], Pedro Lorite[12], María C Vega-Gómez[9], Miriam Rolon[9], Sebastián Pita[1]/+**

[1]Universidad de la República, Facultad de Ciencias, Sección Genética Evolutiva, Montevideo, Uruguay
[2]Universidad de Buenos Aires, Facultad de Ciencias Exactas y Naturales, Departamento de Ecología, Genética y Evolución, Laboratorio de Eco-Epidemiología, Ciudad Autónoma de Buenos Aires, Argentina
[3]Consejo Nacional de Investigaciones Científicas y Técnicas - Universidad de Buenos Aires, Instituto de Ecología, Genética y Evolución, Ciudad Autónoma de Buenos Aires, Argentina
[4]Universidad de Buenos Aires, Facultad de Ciencias Exactas y Naturales, Departamento de Biodiversidad y Biología Experimental, Ciudad Autónoma de Buenos Aires, Argentina
[5]Centro Nacional de Diagnóstico e Investigación en Endemo-Epidemias, Administración Nacional de Laboratorios e Institutos de Salud Dr Carlos Malbrán, Buenos Aires, Argentina
[6]Consejo Nacional de Investigaciones Científicas y Técnicas, Buenos Aires, Argentina
[7]Fundação Oswaldo Cruz-Fiocruz, Instituto Oswaldo Cruz, Laboratório Interdisciplinar de Vigilância Entomológica em Diptera e Hemiptera, Rio de Janeiro, RJ, Brasil
[8]Centro Nacional de Diagnóstico e Investigación en Endemo-Epidemias, Administración Nacional de Laboratorios e Institutos de Salud Dr Carlos Malbrán, Unidad Operativa de Vectores y Ambiente, Córdoba, Argentina
[9]Fundación Moisés Bertoni, Centro para el Desarrollo de la Investigación Científica, Díaz Gill Medicina Laboratoria, Asunción, Paraguay
[10]Universidad Nacional de Catamarca, Centro Regional de Energía y Ambiente para el Desarrollo Sustentable, San Fernando del Valle de Catamarca, Catamarca, Argentina
[11]Universidad Nacional de Córdoba, Facultad de Ciencias Exactas Físicas y Naturales, Instituto de Investigaciones Biológicas y Tecnológicas, Córdoba, Argentina
[12]Universidad de Jaén, Departamento de Biología Experimental, Área de Genética, Jaén, Spain

**BACKGROUND** The conformation of the Sordida subcomplex has been a topic of prolonged debate, with diverse methodological approaches employed to discern its constituent species. Up to now, *Triatoma sordida*, *T. garciabesi* and *T. rosai* comprise part of this subcomplex. Distinguishing and identifying these three species pose significant challenges due to their pronounced morphological similarity, overlapping distributions, and presence of natural hybrids.

**OBJECTIVES** This study aims to uncover the genetic diversity and geographic spread of these three species.

**METHODS** We analysed a mitochondrial *cytochrome b* gene fragment and complemented it with chromosomal studies across natural populations from an extensive geographical range, including Argentina, Bolivia, Brazil, and Paraguay.

**FINDINGS** Phylogenetic analyses revealed genetic distances that suggest the presence of at least six putative species, rather than the three currently recognised.

**MAIN CONCLUSIONS** The present findings underscore the potency and significance of molecular analyses from natural populations for species identification and highlight the limitations of morphology in classifying Triatominae species.

Key words: Chagas disease vectors - cytochrome b - molecular systematics - species delimitation - cytogenetics

The Triatominae subfamily (Hemiptera, Reduviidae) comprises over 154 species of blood-sucking insects that vary in various aspects of their biology, particularly in their significance as vectors of Chagas disease (CD).[1,2,3,4] Among the 18 genera comprising the subfamily, the genus *Triatoma* Laporte, 1832 stands out as the largest with more than 80, categorised into eight complexes and 15 subcomplexes.[4,5] One of these is the Sordida subcomplex, where so far six species have been formally recognised: *Triatoma sordida*,[6] *Triatoma gar-*

Financial support: Programa Desarrollo de Ciencias Básicas (PEDECIBA, Uruguay), Sistema Nacional de Investigadores (SNI-ANII), Consejo Nacional de Investigaciones Científicas y Técnicas de Argentina (CONICET).
GB-R and JN contributed equally to this work.
+ Corresponding author: spita@fcien.edu.uy
 https://orcid.org/0000-0002-4102-5808

ciabesi,[7] *Triatoma rosai*,[8] *Triatoma jurbergi*,[9] *Triatoma matogrossensi* [10] and *Triatoma vandae*.[11] These six species (*T. rosai* was referred as *T. sordida* Argentina) were grouped within the Sordida subcomplex through chromosomal markers,[12] and later confirmed through analysis of nuclear and mitochondrial DNA sequences. [5,13] In 1998, García and Powell, using DNA sequences of mitochondrial genes 12S, 16S, and *cytochrome c oxidase subunit I* (coI), demonstrated that *T. sordida* from Argentina and Brazil form a well-supported clade, which appears as the sister taxon to the *guasayana-rubrovaria-circummaculata* clade.[14] The initial inclusion of *Triatoma guasayana*[15] and *Triatoma patagonica*[16] in this subcomplex, based on morphological similarities,[17,18] has been discarded by numerous chromosomal, molecular and chemical evidences.[5,12,13,19] Enzyme variability in four *Triatoma* species from Argentina, including *T. sordida*, showed that this species exhibited a low level of heterozygosity compared to *T. guasayana*. The results of this study also indicated that *T. sordida* and *T. guasayana* form a closely related species pair.[20] Several authors based on mitochondrial and nuclear sequences, grouped *T. guasayana* within the Sordida subcomplex.[8,21]

Until the revision made by Lent and Wygodzinsky,[1] *T. sordida* constituted a widely distributed species along Argentina, Bolivia, Brazil, Paraguay and Uruguay, showing a great polymorphism. This species, frequently infected with the parasite *Trypanosoma cruzi*,[22] has populations colonising the domicile and peridomicile in some areas, while in other regions, it was observed in a great diversity of wild habitats.[1] Since the 1950s, variations in external colouration and body size have been described among *T. sordida* from Argentina: larger, lighter-coloured domestic individuals in the northeast and smaller, darker sylvatic individuals in the northwest.[23,24] Although unpublished by Wygodzinsky and Abalos, crossbreeding experiments between these two Argentinian morphs demonstrated complete fertility for a minimum of two generations.[24] Actis et al. described variations in the electrophoretic profiles of the haemolymph proteins between sylvatic individuals from northwest Argentina and domestic individuals from Brazil.[25] In 1967, based on chromatic and morphological differences with domestic *T. sordida*, the northwestern Argentinian individuals were described as a new species named *T. garciabesi*,[7] subsequently synonymised with *T. sordida* by Lent and Wygodzinsky.[1] Later, Panzera et al. identified three isoenzymatic groups within *T. sordida*: the northwest and northeast groups from Argentina, and the Brazilian group with three diagnostic loci among individuals from Brazil and Argentina, and one locus between both Argentinian groups.[26] At the chromosomal level both Argentinian groups clearly differed from the Brazilian group, but they were indistinguishable from each other in the amount of autosomal constitutive heterochromatin. Considering the preceding studies, Jurberg et al. revalidated *T. garciabesi* as a distinct species, involving the individuals from northwest Argentina.[27] This revalidation relied on morphological traits (such as overall size, coloration, head, and genitalia morphology) as well as genetic data.

The variability within *T. sordida* also extended to Bolivia, where two distinct putative species were recognised through multilocus enzyme analyses.[28] These groups originally identified as Group 1 and Group 2 (now recognised by chromosomal markers as *T. sordida* and *T. garciabesi* respectively[29] were sympatric in some regions and even interspecific hybrids were identified. More recently, quantitative and qualitative analyses of the cuticular hydrocarbons[30] recognised three different groups within *T. sordida*: *T. garciabesi*, *T. sordida* from Brazil and *T. sordida* from Argentina, similar as described by Panzera et al.[26] In Paraguay, comparative analyses of *T. sordida* populations from western and eastern regions reveal striking differences in the random amplified polymorphic DNA profiles (RAPD), head and wing morphometric and feeding patterns.[31] These authors attribute these differences to population polymorphism caused by eco-geographical isolation by distance.

In 2015, a cytogenetic study involving 139 individuals using 45S rDNA location and C-banding, confirmed the presence of four distinct and well-defined groups within *T. sordida sensu lato (s.l.)*: *T. garciabesi* (samples from Northwest and Central Argentina, western Paraguay and the Bolivian Chaco), *T. sordida sensu stricto (s.s.)* (samples from Brazil, Central and eastern Paraguay, and the Bolivian Chaco), *T. sordida* Argentina (specimens from northeast Argentina, eastern Paraguay, and the Bolivian high valleys) and *T. sordida* La Paz (domestic individuals from the highlands of La Paz, Bolivia).[29] Based also on *coI* sequence data, this study proposed the existence of a new species, widely distributed in the northeast of Argentina (named *T. sordida* Argentina).

Further investigations using individuals from natural populations, analysed morphometric shape measurements for the head, right wing and pronotum, and were able to differentiate *T. sordida* Argentina from *T. sordida* from Brazil (plus Bolivia), as well as from *T. garciabesi*.[32,33] However, both studies (independently conducted and using material from different geographical locations), agreed that a high percentage of individuals from different species overlap in their measurements, which could lead to misidentification of several individuals using these characters. A similar conclusion was obtained with geometric morphometric of the hemelytra.[13] According to Nattero et al. this misidentification can be attributed to various factors, including local adaptation, genetic drift, a high level of phenotypic plasticity, morphological convergence, or even natural hybridisation.[33]

Finally, in 2020, *T. sordida* from Argentina was formally described as a new species, named *T. rosai*. This taxonomic designation was based on the previously mentioned knowledge and new analyses encompassing morphology, crossbreeding, morphometric, and molecular data.[8] The morphological characterisation of this novel species involved a comparative study of individuals from an Argentine population (Corrientes, San Miguel, designated as the holotype for *T. rosai*) and individuals of *T. sordida* s.s. from Brazil (Minas Gerais).

Although there are morphological keys that allow the identification of *T. sordida*, *T. garciabesi* and *T. rosai*, their practical application still remains as a challeng-

ing task. The recognition and differentiation of each of these three species have been and continue to be subject to controversies due to their high morphological similarity, partially overlapping geographical distributions, and even the existence of natural hybrids among them. Molecular identification and evolutionary relationships among these species and with *T. guasayana* have also proven to be highly problematic due to contradictory results. For example, phylogenetic trees based on a *coI* fragment showed that *T. sordida*, *T. garciabesi* and *T. rosai* form a monophyletic group, with *T. guasayana* as an outgroup.[29] Similar results were reported using concatenated mitochondrial and nuclear genes: *coI, cytochrome b* (*cytb*), 16S, 18S, and 28S.[13] However, a study from the same research group that employed nearly identical markers and sequences (*coI, cytb*, 16S, 28S, and ITS-2) placed *T. guasayana* within the same clade as *T. sordida* and its closely related species.[8]

All these species also differ in their epidemiological importance as vectors of CD. *T. sordida* exhibits significant epidemiological variation across its distribution range in terms of infection rates with *T. cruzi* and its ability to colonise domestic and peridomestic habitats. This variation has been documented in different countries, including Brazil,[32,34,35,36] Bolivia,[37] and Paraguay.[38,39] *T. rosai* populations were primarily found in chicken coops, showing a limited ability to colonise and a low rate of infestation in human dwellings.[40,41] *T. garciabesi* has been found in nests of Furnariidae and Psittacidae birds, with few records in peridomestic environments.[42,43,44]

This study focuses on phylogenetic and population genetic analyses using *cytb* sequences of *T. sordida*, *T. garciabesi*, and *T. rosai*, including not only new sequences from several natural populations from Argentina, Bolivia, Brazil and Paraguay, but also the complete dataset of sequences reported as *T. sordida* and related species deposited in GenBank. We also included *T. guasayana*, a species often mistaken with *T. sordida*, due to their significant morphological resemblance and overlapping geographic ranges.[45] These *cytb* studies were complemented, whenever possible, with analyses on the same individuals using previously established chromosomal markers that differentiate the aforementioned species. The aim of this study is to elucidate the molecular differentiation among these species and properly establish their geographical distribution ranges. Given the variability in vector competence and the frequent taxonomic confusion among these species, a more accurate understanding of their geographical distribution could enhance the assessment of CD transmission risk in the Southern Cone of South America.

## MATERIALS AND METHODS

*Insects and collection sites* - A total of 95 male adults from natural populations of 43 localities from Argentina, Bolivia, Brazil and Paraguay [Supplementary data (Table I, Fig. 1)] were studied. Some of these samples — 25 individuals — were also analysed by the chromosomal markers detailed in Panzera et al.[29] (depicted in red in Figure). All insects were collected from peridomestic structures,

except intradomestic individuals from Inquisivi (La Paz, Bolivia) and sylvatic ones from Reserva Natural y Cultural Bosques Telteca (Mendoza, Argentina). Using the available morphological keys,[1,8,27] all specimens were identified as members of the Sordida subcomplex.

*DNA extraction and sequencing* - For DNA extraction, three legs preserved in 70% ethanol were used from each specimen and total DNA was extracted by a standard phenol-chloroform procedure. The mitochondrial *cytb* gene was selected for the analyses as it has been proven to be successful for the study of other species complexes within Triatominae,[46] and because it is the mitochondrial fragment with the highest number of sequences available in the GenBank. A fragment of about 500 bp was amplified by polymerase chain reaction (PCR) using primers CYTB7432F[46] and Cob-82R-DEGEN.[47] Amplifications were generated by 30/35 cycles of 30 s at 95ºC, 30 s at 58/47ºC and 1 min at 72ºC, preceded by 5 min at 95ºC and followed by 10 min at 72ºC. The PCR products were sent to Macrogen Inc. (Korea) for DNA purification and sequencing. Sequences were manually curated by chromatogram evaluation using Chromas (https://technelysium.com.au/wp/chromas/), and then deposited in the GenBank database (http://www.ncbi.nlm.nih.gov), under accession numbers PP972075 to PP972104 [Supplementary data (Table II)].

*DNA sequence analyses* - Given the high morphological similarity among *T. sordida*, *T. garciabesi*, *T. rosai*, and even with *T. guasayana*, we conducted a NCBI search of all GenBank sequences identified as *T. sordida*, *T. garciabesi*, and *T. guasayana*. This strategy allowed us to recognise potentially misidentified bugs, based on morphological characters. Both sequenced individuals of *T. guasayana* (PP972075 and PP972076) were identified as belonging to this species according to the chromosomal markers described by Panzera et al.[29]

Despite sequenced a fragment of approximately 500 bp, genetic analyses were performed using a 233 bp fragment, in order to include a larger number of the GenBank sequences. Several sequences of great importance deposited in GenBank (*i.e.*, sequences KR822185 to KR822199) were short, and only included the central region of our dataset. The same issue was previously experienced by us analysing a dataset of *cytb* of other species, *Triatoma maculata*,[48] where short sequences deposited in GenBank were extremely important to leave them aside. Therefore, analyses were run for the long and short fragments. Those results demonstrated that both fragments retrieved the same information.[49] As an outgroup taxon, we selected *T. rubrovaria*,[50] a species morphologically well differentiated from Sordida subcomplex species but belonging to an evolutionarily close subcomplex — the same as *T. guasayana*. Sequence alignment was performed using MAFFT v7.310.[51]

The R core base version 3.6.1,[52] and R package *phangorn*[53] were used to estimate a phylogenetic maximum likelihood (ML) tree with 1000 bootstrap pseudoreplicates for node support. Bayesian Inference (BI) was implemented in MrBayes3.2.7.[54] For BI we used four Markov chains for 12 runs of 20 million iterations.

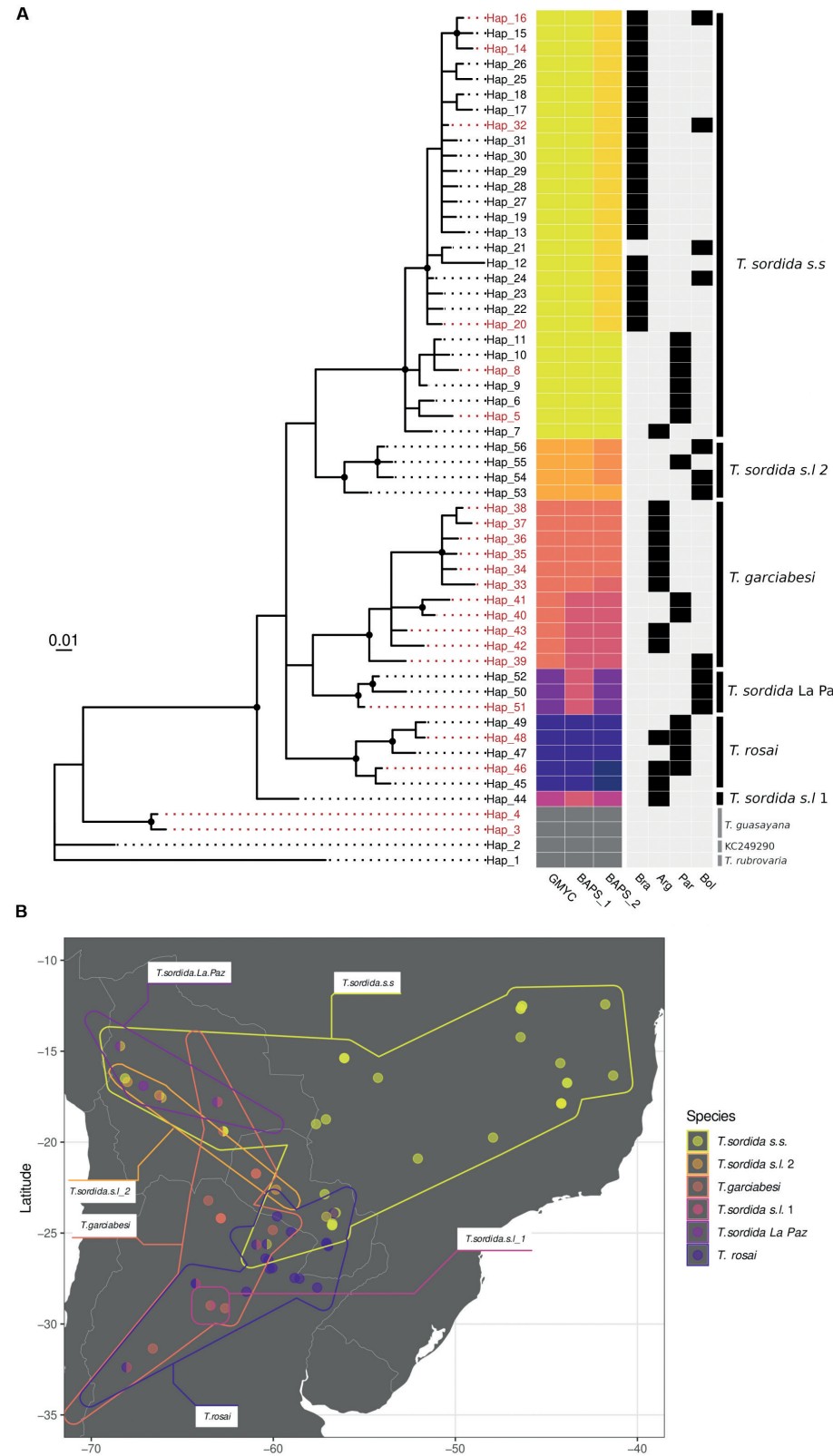

(A) Bayesian inference phylogenetic tree obtained from cytochrome b fragment (233 bp). Posterior probabilities: support is depicted with black dots over the nodes when above 0.95. In red are the haplotypes that have been cytogenetically analysed. Colours on the generalised mixed yule coalescent (GMYC) column define the mitochondrial lineages and used in part B and in Supplementary data (Fig. 3). The two columns on the right represent results from hierBAPS algorithm, different tones of the same colours mean discrepancy with GMYC. The dark grey colour represents the outgroup. The country distribution is represented by black squares on the light grey columns. (B) Map from the Southern Cone in South America, indicating the geographical distribution of six clades. When two clades are in the same locality, the circle is splitted in two.

The JModeltest function implemented in the *phangorn* package was used to find the best-fitted nucleotide substitution model, and decisions were taken under the Bayesian Information Criterion (BIC). The phylogenetic trees were visualised and edited in the R package *ggtree*.[55] Next, clades within the dataset were defined employing Hierarchical Bayesian Analysis of Population Structure (HierBAPS) implemented in the R package *rhierbaps*.[56,57] In order to have an integrated view of putative species delimitation, the Generalised Mixed Yule Coalescent (GMYC) method[58] was applied. For this purpose, an ultrametric tree was employed, which was based on a Yule process and obtained using BEASTv.2.3.7.[59] GMYC was carried out in R using *ape*[60] and *splits* packages.[61] The length of the MCMC chain was 100 million iterations sampling every 10,000. Stationarity and convergence of all parameters was verified using the Tracer v 1.5 software.[62]

The R core base version 3.6.1,[52] *vegan* (https://CRAN.R-project.org/package=vegan), *ape*[60] and *pegas*[63] packages were employed to calculate the within and intergroup pairwise Kimura two parameters (K2p) distance, Tajimas' D test[64] as well as nucleotide diversity (π), number of haplotypes (h), segregating sites (S) and haplotype diversity (H). In addition, the MMD function of the *pegas* package was used to draw a histogram of the frequencies of pairwise distances for each of the three largest lineages from the *cytb* dataset (*T. sordida s.s.*, *T. garciabesi* and *T. rosai*), which represent the mismatch distribution plot. The *adegenet* and *hierfstat* R packages were utilised to calculate Fst statistics. Finally, we construct a minimum spanning tree (MST) haplotype network using PopART software.[65]

*Chromosomal studies* - Whenever possible, to complement molecular identification, individuals were also analysed using chromosomal markers. The chromosomal identification criteria for each species included chromosomal position of ribosomal clusters (by fluorescent *in situ* hybridisation) and distribution of C-heterochromatin (by C-banding), as previously described in Panzera et al.[29]

## RESULTS

The total analysed dataset included 56 haplotypes corresponding to *T. sordida*, *T. garciabesi*, *T. rosai*, *T. guasayana* and *T. rubrovaria* [Figure, Table I, Supplementary data (Table II)]. Among these, 31 haplotypes correspond to this study, while the other 25 haplotypes were sequences extracted from GenBank [Supplementary data (Table II)].

GMYC and hierBAPS algorithms retrieved similar results from each other. Differences between GMYC and hierBAPS level 1 involved *T. garciabesi*, *T. sordida* La Paz and *T. sordida sensu lato* 1 (*T. sordida s.l.* 1). These are resolved if hierBAPS level 2 is considered. The exception is that *T. garciabesi* is split into two groups (depicted as change in hue colour in the column corresponding to hierBAPS level 1 within the right panel of Figure A). This could be reflecting a division in two lineages within *T. garciabesi* as described recently.[66] A deeper sampling effort, focused on *T. garciabesi* must be performed in order to evaluate the hypothesis. Therefore, GMYC results were taken as the best option to describe the phylogenetic groups or clades.

According to our analysis, seven GenBank sequences were misidentified [Supplementary data (Table I, in red)]. Surprisingly, a *T. sordida* GenBank sequence (Hap. 1 — KC249290) was also included within the outgroup, presenting a high identity (98.71%) with *T. infestans*[67] (Acc. Number HQ333230). Two *T. sordida* GenBank sequences (KC249293 and KC249295) were identified as *T. garciabesi* and *T. rosai*, respectively. Finally, other four GenBank sequences erroneously identified as belonging to *T. guasayana* grouped into two clades within the Sordida subcomplex (*T. sordida s.s.* — Hap. 24: KC249250 — and *T. sordida* La Paz — Hap. 50: MH054944; Hap. 52: KC249252 and KC249253) [Figure A, Supplementary data (Fig. 1)].

According to the topology of the BI (Figure A) and ML trees [Supplementary data (Fig. 1)] we recognised six clades within the Sordida subcomplex. In the ML tree all clades represent a basal polytomy, hence phylogenetic relationships at the base of the tree were not resolved

TABLE I

Mitochondrial DNA sequence diversity among the six clades of Sordida subcomplex identified by phylogenetic analyses plus *Triatoma guasayana* as an outgroup. Estimates derived from analysis of *cytochrome b* fragment (233 bp)

| Species | N | Nh | S | Hd | pi | Tajimas' D | Tajimas' D p-value |
|---|---|---|---|---|---|---|---|
| *T. sordida sensu stricto* | 160 | 28 | 32 | 0.706 | 0.007 | -2.132 | 0.033 |
| *T. garciabesi* | 22 | 11 | 25 | 0.909 | 0.027 | -0.371 | 0.711 |
| *T. sordida sensu lato* 1 | 1 | 1 | 0 | NA | NA | NA | NA |
| *T. rosai* | 48 | 5 | 8 | 0.368 | 0.007 | -0.167 | 0.867 |
| *T. sordida* La Paz | 4 | 3 | 4 | 0.833 | 0.009 | -0.780 | 0.435 |
| *T. sordida sensu lato* 2 | 4 | 4 | 7 | 1.000 | 0.015 | -0.817 | 0.414 |
| *T. guasayana* | 2 | 2 | 1 | 1.000 | 0.004 | NA | NA |

N: number of specimens analysed; Nh: number of haplotypes; S: number of segregating sites; Hd: haplotype diversity; π: nucleotide diversity; Tajimas' D neutrality test value and its p-value.

[Supplementary data (Fig. 1)]. In the BI tree, a similar basal polytomy is found after the divergence of the Hap. 44, which belongs to *T. sordida s.l.* 1 (Figure A).

In Figure, the first clade from top to bottom (depicted in yellow — haplotypes 5 to 32) corresponds to *T. sordida s.s.* and is located mainly in Brazil, but also in Bolivia, Paraguay and one haplotype (Hap. 7) in Argentina [detailed also in Supplementary data (Fig. 2)]. The second clade (in pale orange — haplotypes 53 to 56), corresponds to *T. sordida sensu lato* 2 (*T. sordida s.l.* 2), distributed in Bolivia and Paraguay. The third clade (coloured in dark orange — haplotypes 33 to 43), is *T. garciabesi*, located mainly in Argentina but also in Paraguay and Bolivia. The fourth clade (coloured in light purple, haplotypes 50 to 52), corresponds to *T. sordida* La Paz*,* which is distributed exclusively in Bolivia. The fifth clade, (dark purple — haplotypes 45 to 49), is *T. rosai* (*T. sordida* Argentina *sensu* Panzera et al.),[29] which is distributed mainly in Argentina, but also in Paraguay. The last clade at the bottom of the in-group (séance colour) possesses only haplotype 44 located in Argentina, and receives the name of *T. sordida s.l.* 1. In Supplementary data (Fig. 2), the detailed geographical distribution of each haplotype is shown, discriminated by clades with the same colours as in Figure.

All clades retrieved by GMYC and hierBAPS algorithms are clearly represented in the MST network [Supplementary data (Fig. 3)]. It could be seen how mutational steps are larger in the branches that separate each group (from 11 to 14 between groups compared to a maximum of seven within a group). It is important to note that the haplotype 44 (a singleton of *T. sordida s.l.* 1) is separated from the other haplotypes by several mutational steps [Supplementary data (Fig. 3)]. The star-like shape of the haplotype network for *T. sordida s.s.* is also particular, featuring a central variant encircled by haplotypes exhibiting only minor mutational differences compared to the structure observed in other species such as *T. garciabesi*.

The greatest number of haplotypes (28) and segregating sites (32) was found in *T. sordida s.s.* (Table I). However, the highest haplotype diversity was in *T. garciabesi* (0.91) and *T. sordida s.l.* 2 (1.0) and the lowest in *T. rosai* (0.37). The highest nucleotide diversity value was 0.027 in *T. garciabesi*, and the lowest 0.007 in *T. sordida s.s.* and *T. sordida* La Paz. Tajimas' D was significant only for *T. sordida s.s.* (-2.132, p = 0.03) (Table I), therefore indicating population size expansion or a selective sweep. The mismatch distribution plots over the most sampled lineages are showing two different patterns. For *T. sordida s.s.* and *T. rosai* a skewed histogram is seen where most individuals present similar haplotypes, with just a few divergent ones. On the other hand, *T. garciabesi* presents a bimodal plot [Supplementary data (Fig. 4)].

According to K2p corrected genetic distances (Table II), a *T. sordida* GenBank sequence (Hap. 1 — KC249290) displays genetic distances higher than 21% with any of the Sordida subcomplex species. The smallest genetic distance observed for haplotype 1 (erroneously identified as *T. sordida*, likely *T. infestans*) was 16% with *T. rubrovaria*. Regarding the two-outgroup species, *T. rubrovaria* and *T. guasayana*, they display

## TABLE II

Mean Kimura 2-parameter (K2p) distances (expressed in percentages) of *cytb* sequences among the clades of Figure, plus *Triatoma sordida* GenBank (Hap. 1 - KC249290), *T. rubrovaria* and *T. guasayana* as outgroups. Mean within-clade distances are in bold on the diagonal

| | Hap. 1 KC249290 | T. rubrovaria | T. guasayana | T. garciabesi | T. rosai | T. sordida La Paz | T. sordida s.l. 1 | T. sordida s.l. 2 | T. sordida s.s. |
|---|---|---|---|---|---|---|---|---|---|
| T. rubrovaria | 15.99 | **NA** | | | | | | | |
| T. guasayana | 17.39 | 6.63 | **0.43** | | | | | | |
| T. garciabesi | 23.35 | 13.88 | 13.35 | **3.25** | | | | | |
| T. rosai | 23.14 | 15.04 | 16.54 | 10.09 | **2.02** | | | | |
| T. sordida La Paz | 22.43 | 14.07 | 12.26 | 7.09 | 7.60 | **1.16** | | | |
| T. sordida sensu lato 1 | 21.07 | 12.75 | 12.49 | 6.97 | 6.34 | 6.05 | **NA** | | |
| T. sordida sensu lato 2 | 22.87 | 13.91 | 14.99 | 7.82 | 7.16 | 6.11 | 6.50 | **1.54** | |
| T. sordida sensu stricto | 24.62 | 14.58 | 18.99 | 10.61 | 8.37 | 9.91 | 9.44 | 8.34 | **2.04** |

genetic distances ranging from 12% to 19% when compared to Sordida subcomplex species. Within the Sordida subcomplex, the greatest distance was between *T. sordida s.s.* and *T. garciabesi* (10.61%), and the smallest was 6.05% (seen between *T. sordida s.l.* 1 versus *T. sordida* La Paz). The intra-clade variation ranges between 1.16% and 3.25%, as seen in *T. sordida* La Paz and *T. garciabesi*, respectively (Table II). Genetic distances results are also given for haplotypes on Supplementary data (Table II). In addition, Fst statistics revealed great differentiation between lineages, with values ranging from 0.78 to 0.96.

Despite that we obtained results for new *T. sordida s.s*, *T. garciabesi* and *T. rosai* individuals of locations not studied earlier, chromosome analyses showed the same patterns already described in Panzera et al.[29] In *T. sordida s.s*, autosomal C-heterochromatin was found in several autosomal pairs and 45S ribosomal clusters on the X chromosome. *T. garciabesi* had no autosomal C-heterochromatin and ribosomal clusters on the X chromosome. *T. rosai* showed no autosomal C-heterochromatin and 45S ribosomal clusters present on both XY sex chromosomes.

## DISCUSSION

The Sordida subcomplex has long posed taxonomic challenges due to the morphological similarity among species, their overlapping distributions, and evidence of natural hybridisation. To contribute to a clearer understanding of this group, we analysed a mitochondrial gene (*cytb*) fragment and chromosomal markers across 43 new populations from Argentina, Bolivia, Brazil, and Paraguay. Our results suggest the existence of at least six genetically distinct lineages — a diversity greater than currently recognised — and highlight regional patterns that were not evident from morphology alone. Based on a broad geographic sampling, this investigation report evidence of an extensive genetic diversity involving *T. sordida* and related species, leading to the following conclusions:

*Phylogenetic groups* - Within the Sordida subcomplex species, we recognise six clades with pairwise *cytb* K2p distances greater than 6%, and with support values (BPP) higher than 0.90 [Table I, Supplementary data (Fig. 1)]. The three clades formally recognised as true species (*T. sordida s.s.*, *T. garciabesi* and *T. rosai*) showed genetic distances between 8.4% to 10.6%, very similar as those reported by Alevi et al.[8] (7.4% to 9.7%). The other three clades, *T. sordida s.l.* 1, *T. sordida s.l.* 2, and *T. sordida* La Paz, exhibit genetic distances between 6-11%. Studies of DNA barcoding and DNA taxonomy have proposed that threshold values of sequence divergence can assist in delineating animal species.[68] Anyway, the divergence thresholds are arbitrary and can vary widely depending on the genetic marker employed and even among different insect groups.[69] Mitogenome analyses of 90 hemipteran species indicated that *cytb* and *coI* gene fragments undergo evolution at comparable rates.[70] However, within triatomines, *cytb* gene evolves slightly more rapidly than *coI*.[71,72] In *Triatoma* sister species, *cytb* sequence divergence levels (measured by K2p distances) generally exceed 7.5%,[71,73,74] significantly surpassing the 2-3% threshold commonly used in several insect groups, including true bugs.[69,75] In our analyses, *cytb* K2p distances range from 6% to 11% among the six clades here identified, whereas within-clades distances do not exceed 3.3% (Table II). Although the three clades not recognised as valid species fall just below the 7.5% threshold (*T. sordida* La Paz, *T. sordida s.l.* 1, and *T. sordida s.l.* 2), these results suggest strong genetic differentiation within the species now recognised as *T. sordida*. If we compare these three taxa exclusively with *T. sordida s.s.*, the genetic distances are always greater than 8.3% (range 8.3% to 9.9%). More extensive phylogenetic analyses with various nuclear and mitochondrial markers could clarify whether these three clades involve new species or not.

*Misidentified species sequences* - This work highlights the incorrect taxonomic identification of numerous *cytb* sequences from GenBank, as it has been reported for other mitochondrial DNA sequences[13,29] [Figure, Supplementary data (Table III)]. The most striking example is the haplotype 1 identified as *T. sordida* (KC249290), with such a high level of molecular divergence that it even appears as an outgroup in our analyses (Figure, Table II). Regarding *T. guasayana*, the four GenBank sequences named would not belong to this species. Furthermore, *T. guasayana* from GenBank did not cluster with individuals previously identified as *T. guasayana* based on chromosomal characteristics reported by Panzera et al.[29] — absence of autosomal heterochromatin and ribosomal DNA clusters on a single autosomal pair. These incorrect identifications in GenBank accessions explain the contradictory results of numerous reports whether *T. guasayana* is within or outside the Sordida subcomplex[8,76] as commented in the "Introduction" section.

*Geographical distribution and genetic variability within each clade. T. sordida sensu stricto* - This species is widely distributed in Brazil, Paraguay, Bolivia (highlands and Chaco region). One specimen from Argentina (Chaco, El Colchon, Hap. 7) (Figure) is included but could be a case of introgression. Furthermore, it is sympatric with the haplotype 48 of *T. rosai*. *T. sordida s.s.* also possess overlapped distribution with clades *T. garciabesi*, *T. sordida* La Paz, and *T. sordida s.l.* 2 in Bolivia. All the sordida-like specimens found in Brazil belong to this clade [Supplementary data (Fig. 2A)]. Several haplotypes were confirmed by chromosomal markers (identified in red in Figure). Despite its widespread geographic distribution and the large number of individuals analysed, along with a high number of haplotypes and segregating sites (Table I), this species exhibits the lowest nucleotide diversity among the six identified clades, with K2p interclade distances at 2.0% (Table II). This narrow molecular variability aligns with previous isoenzymatic and DNA studies.[77,78,79] The K2p genetic distances of *T. sordida s.s.* with the remaining five clades are greater than 8.3%, including the three taxa not formally recognised as species (Table II). A significant negative Tajima's D value together with high haplotype diversity and low nucleotide diversity, the unimodal

mismatch distribution plot [Supplementary data (Fig. 4)] and the star-shaped haplotype network [Supplementary data (Fig. 3)] for this species suggest a recent process of population expansion, indicative of a strong colonising capability in this taxon relative to others within the subcomplex. The same individuals previously identified as chromosomal hybrids (due to heterozygous 45S cluster marking in an autosomal pair, leading to the hypothesis of *T. sordida* La Paz with other species) by Panzera et al.[29] from Izozog (Bolivia) were examined and found to have *T. sordida s.s. cytb* haplotypes (Hap. 16 and 32).

*Triatoma sordida sensu lato 1* - The clade is restricted to the sequence of one individual (Hap. 44), separated from the other haplotypes by several mutational steps [Supplementary data (Fig. 3)], and localised in the dry Chaco region from Argentina (Santiago del Estero, Salavina) [Supplementary data (Fig. 2D)]. Other individuals from the Sordida subcomplex that are geographically closest are identified as *T. garciabesi*, with genetic distances approximately 7% (Table II). It is noteworthy that this individual shows a considerable genetic distance from *T. sordida s.s.* (9.44%), indicating that it clearly represents a different taxon. *T. sordida s.l.*1 is also distant to *T. sordida* La Paz and *T. sordida s.l.* 2 (K2p distances of 6.1% and 6.5%, respectively), which are geographically very distant [Supplementary data (Fig. 2)]. Chromosomal analyses were not possible to carry out for this taxon either.

*Triatoma sordida sensu lato 2* - The clade is distributed in Paraguay (Boquerón Department) (haplotype 55), and three GenBank sequences from highlands of Bolivia (Hap. 53, 54 and 56) [Supplementary data (Fig. 2B)]. In Bolivia, this taxon occurs in the same geographic areas as *T. sordida s.s.* and *T. sordida* La Paz, from which it is separated by K2p distances of 8.3% and 6.1%, respectively. Chromosomal analyses were not possible to perform for this taxon, as their gonads were not collected while the insects were still alive.

*Triatoma garciabesi* - This species is common and widely distributed in northern Argentina, although it is also found in a locality in Paraguay (Boquerón, haplotypes 40 and 41) and in the Bolivian Chaco (Hap. 39) [Figure, Supplementary data (Fig. 2D)]. In several localities of Argentina, this species is found in the same areas as *T. rosai*. This clade is characterised by presenting the highest nucleotide and haplotype diversity among the six clades analysed (Table I), with an intraclade K2p distance of 3.3% (Table II). Moreover, the bimodal distribution on the mismatch distribution plot is consistent with a marked structured population [Supplementary data (Fig. 4)], as the presence of sequences separated by multiple mutational steps in the haplotype network [Supplementary data (Fig. 3)]. This high intraspecific variation in *cytb* closely remembers the pattern recently found in *coI* fragments.[66] The eleven haplotypes identified in this species were chromosomally analysed (Figure).

*Triatoma rosai* - This study reports this species in several provinces of Argentina (some newly reported) and departments of Paraguay [Figure, Supplementary

data (Fig. 2E, Table I)]. Chromosomal and cuticular hydrocarbon profiles have identified the presence of this taxon also in Bolivia in Cochabamba and Santa Cruz Departments.[19,29] Despite this wide distribution, the species exhibits a low number of haplotypes with the lowest nucleotide diversity (Table I). Several individuals of this species (haplotypes 46 and 48) were examined, and corroborated via chromosomal markers (Figure A).

*Triatoma sordida La Paz* - This taxon is found exclusively in Bolivia (La Paz, Cochabamba and Santa Cruz) and comprises three haplotypes (50, 51, and 52) [Figure, Supplementary data (Fig. 2E)]. Two of them are retrieved from GenBank and classified as *T. guasayana*. The third one, Hap. 51, described here from an individual that was chromosomally analysed in Panzera et al.,[29] presents 45S ribosomal clusters in an autosomal pair and C-heterochromatic autosomes, distinctive traits of *T. sordida* La Paz. Our analysis places these three haplotypes within the same clade, which is genetically quite distinct from *T. guasayana* (K2p = 12.3%) and *T. sordida s.s.* (9.9%).

A recent research proposed that *T. sordida* La Paz was not a valid species due it low genetic distance with *T. sordida s.s.*[80] To verify this conclusion, the same *cytb* sequences from Madeira et al.[80] were used here and confirmed that these individuals of *T. sordida* from the La Paz region are in fact *T. sordida s.s.* (haplotype 32 = MZ700100 and haplotype 21 = MZ700101). Madeira et al. failed to assume that all *T. sordida* individuals from La Paz region belonged to the lineage *T. sordida* La Paz.[80] Our molecular and chromosomal data indicate the coexistence of both clades, *T. sordida s.s.* and *T. sordida* La Paz, in the La Paz region, with significant genetic distinctions between them (K2p distances of approximately 10%). This substantiates the need for an exhaustive morphological analysis of *T. sordida* La Paz to assess its status as a distinct species. Cuticular hydrocarbon studies on individuals from Cochabamba also suggest the existence of other species beyond those already described,[19] possibly *T. sordida* La Paz or *T. sordida s.l.* 2, as proposed here.

*Integrative taxonomy and evolutionary divergence in the Sordida subcomplex* - Building on the analysis of a mitochondrial *cytb* gene fragment supplemented by chromosomal analyses, our study identifies six molecular taxa within the Sordida subcomplex. Three of these taxa are formally recognised as species (*T. sordida s.s.*, *T. garciabesi*, and *T. rosai*), while the others (*T. sordida s.l. 1*, *T. sordida s.l. 2*, and *T. sordida La Paz*) require further investigation to confirm their taxonomic status. Notably, four of these clades also exhibit chromosomal differences previously described by Panzera et al,[29] underscoring the complexity of this group.

Our study not only provides a detailed molecular and chromosomal framework for understanding the Sordida subcomplex but also emphasises the critical role of integrating multidisciplinary approaches to resolve taxonomic ambiguities. The identification of six mitochondrial lineages — three of them currently undescribed — suggests a more complex evolutionary history than previously recognised. The distinct DNA sequence parameters

(haplotype and nucleotide diversity, neutrality), mismatch distribution patterns, together with high Fst values and phylogenetic support, point to contrasting demographic scenarios and limited gene flow among lineages, consistent with ongoing or past speciation processes. These results also hint at possible historical biogeographic patterns of isolation and expansion across the Gran Chaco, Andean, and Atlantic forest regions. Misidentifications in public databases and the occurrence of natural hybrids further complicate species delimitation, reinforcing the need for robust integrative taxonomic frameworks. Although the limited resolution of mitochondrial data prevents a definitive evolutionary reconstruction, our findings allow us to propose a working hypothesis for the differentiation of the Sordida subcomplex. The expansion signals observed in *T. sordida s.s.* and *T. rosai* may reflect recent colonisation events following ecological or anthropogenic changes, whereas the deep genetic structure and bimodal mismatch pattern in *T. garciabesi* suggest a different evolutionary history, potentially involving incipient speciation or historical fragmentation. Together, these patterns support the view that the Sordida subcomplex comprises lineages at different stages of divergence, shaped by heterogeneous demographic and biogeographic processes across South America. A full understanding of these dynamics will require broader sampling and the incorporation of nuclear genomic data. Finally, we propose the Sordida subcomplex as a promising model system to study speciation, hybridisation, and biogeographic diversification in CD vectors.

Addressing these challenges requires significant collaboration among research groups, leveraging advanced genomic tools such as single nucleotide polymorphism (SNP) analyses and comprehensive chromosomal studies. This approach could not only enhance our understanding of genetic flow and hybridisation but also serve as a model for resolving taxonomic ambiguities in other insect vectors. Moreover, these efforts must overcome practical barriers, including the extensive geographic overlap of clades in regions like the Andean and Gran Chaco areas, and the humid Chaco regions of Paraguay and Argentina, where the collection of samples is logistically and financially demanding.

Our findings also raise intriguing questions about the evolutionary forces driving diversification within the Sordida subcomplex. Future studies could explore whether environmental gradients, host preferences, or historical biogeographic events contribute to the genetic differentiation observed among clades. Conducting comparative analyses of mitochondrial and nuclear markers, alongside chromosomal and SNP data, will be essential to further elucidate the taxonomy of this complex.

## ACKNOWLEDGEMENTS

To EB Oscherov, A Gonzalez and R Cardozo for providing biological materials used in this study.

## AUTHORS' CONTRIBUTION

JN, FP and SP conceived and designed the study; CML, PAL, ARA, MJC, CSR, MCVG, ALCF and MR collected bugs in the field; GBR, NR and SP performed laboratory work and *in silico* analyses; GBR, JN, NR, RVP, ALCF, PL, FP and SP drafted the manuscript; PL, FP and SP funding acquisition. All authors read and approved the final version of the manuscript. The authors declare no conflicts of interest. All sequences used in this article were deposited in GenBank Nucleotide Database (https://www.ncbi.nlm.nih.gov/genbank/) with accession numbers from PP972075 to PP972104.

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

# OPEN PEER REVIEW

Memórias do IOC thanks the anonymous reviewers for their contribution to the peer review of this work.

**FIRST REVIEW ROUND**

REVIEWERS' COMMENTS

### REVIEWER #1

General comments

The manuscript "Reconstructing Sordida subcomplex (Hemiptera, Reduviidae, Triatominae) phylogeny across species distribution range", authored by Gabriela Burgueño-Rodríguez and corresponding author Sebastián Pita, presents a comprehensive phylogenetic analysis of 95 male adult specimens from natural populations across 43 localities in Argentina. The study combines DNA sequence analyses, fluorescent in situ hybridization, and C-heterochromatin distribution (C-banding), while also revisiting previously published data and GenBank entries. A particularly valuable aspect of this work is the identification of potential misassignments in public sequence databases—an important contribution that may help prevent the replication of such errors in future research. The methodology is sound, and the results are clearly presented and thoroughly discussed. Overall, this is a well-executed and rigorous study, well within the scope of Memórias do Instituto Oswaldo Cruz. I recommend its publication with minor revisions. A few suggestions to improve the taxonomic and literature review sections are included as specific comments.

Specific comments

1. To provide a more complete historical context of the evolutionary systematics of the Sordida subcomplex, the authors should consider including key pioneering studies such as: García BA, Canale DM, Blanco A. Genetic structure of four species of Triatoma (Hemiptera: Reduviidae) from Argentina. J Med Entomol. 1995; 32(2): 134-7. doi: 10.1093/jmedent/32.2.134, and García BA, Powell JR. Phylogeny of species of Triatoma (Hemiptera: Reduviidae) based on mitochondrial DNA sequences. J Med Entomol. 1998; 35(3): 232-8. doi: 10.1093/jmedent/35.3.232.

2. The first paragraph of the Discussion (line 387) would benefit from a brief summary of the study's aims and analytical approaches, to guide the reader and establish a clearer narrative transition from the results. As currently written, the section begins abruptly with conclusions, without restating the objectives or the rationale for the study.

3. The incorrect molecular species identification section raises an important point by discussing potentially misidentified cytb sequences in GenBank. While a few accession numbers are mentioned in the main text (e.g., KC249290, KC249293, and KC249295), others are referenced more generally (e.g., "four T. guasayana sequences") without explicitly listing their accession numbers or providing supporting details for reclassification. Although these sequences are noted as highlighted in Supplementary Table S1, the authors might consider including a concise table — either in the main text or as supplementary material—summarizing: (i) the original GenBank accession, (ii) the initial species designation, (iii) the reassigned identity based on the current analyses, and (iv) brief supporting rationale. This addition could improve clarity and reproducibility for readers, though its inclusion is entirely at the authors' discretion.

### REVIEWER #2

This MS represents an important and relevant study to reconstruct the Sordida subcomplex of species belonging to the Triatominae subfamily (Hemiptera, Reduviidae) being most of these taxa vectors of Chagas disease in South America. The initial hypothesis to test for the taxa integration of this subcomplex was correctly achieved. The abstract includes major findings of the present study. This research includes multidisciplinary approaches like molecular systematics, several parameters of population genetics based on Cytb mitochondrial gene, cytogenetics data and biogeography, including important number of samples collected among different location and countries where these taxa distribute in South America. Methodologically is a well conducted research, including robust phylogenetic analyses to support possible scenarios of differentiation for these species subcomplex. In this regard, my major concern refers to this topic. In my modest opinion, the authors could to include other additional analyses (i.e. Mistmach distribution, indirect estimate of gene flow, among others). Thus, the Discussion could be include possible scenarios of taxa differentiation, a historical biogeography hypothesis (patterns and processes) based in this robust analyses (or additional as the aforementiod ones) for the described and undescribed taxa included in this subcomplex. I think that to extend the objective of present paper including and discussing differentiation hypothesis of the Sordida subcomplex would improve it, as potential evolutionary model. Figures and tables, as well references, have the required information to interpret results, but the authors must to check the correct present format for them.

I have made more detailed comments (see in the attached MS revised pdf file and please display all corresponding notes), so I prefer do not repeat here.

Finally, I think that the authors must include the recommendations above mentioned, to obtain more conclusive scenarios of the differentiation for this species group to support biodiversity studies, in this interesting and important group.

## AUTHORS' RESPONSE TO THE REVIEWERS

Dr Adeilton Alves Brandão

Editor in chief

Dr Adeilton Brandão

Handling Editor

Memórias do Instituto Oswaldo Cruz

Thank you very much for the opportunity to resubmit our manuscript. We have revised the reviewers' comments. We are very grateful for the constructive comments. We have taken all the comments seriously and have made as many suggested changes as possible.

We have tried to respond adequately to all the issues raised by both reviewers.

Thank you for your time and consideration. We look forward to hearing from you.

Sincerely,

Sebastián Pita, Ph.D.

Below we respond to each of the reviewer's comments and the modifications carried out in the revised manuscript:

Reviewer #1

General Comments

The manuscript "Reconstructing Sordida subcomplex (Hemiptera, Reduviidae, Triatominae) phylogeny across species distribution range", authored by Gabriela Burgueño-Rodríguez and corresponding author Sebastián Pita, presents a comprehensive phylogenetic analysis of 95 male adult specimens from natural populations across 43 localities in Argentina. The study combines DNA sequence analyses, fluorescent in situ hybridization, and C-heterochromatin distribution (C-banding), while also revisiting previously published data and GenBank entries. A particularly valuable aspect of this work is the identification of potential misassignments in public sequence databases—an important contribution that may help prevent the replication of such errors in future research. The methodology is sound, and the results are clearly presented and thoroughly discussed. Overall, this is a well-executed and rigorous study, well within the scope of Memórias do Instituto Oswaldo Cruz. I recommend its publication with minor revisions. A few suggestions to improve the taxonomic and literature review sections are included as specific comments.

Specific comments

1. To provide a more complete historical context of the evolutionary systematics of the Sordida subcomplex, the authors should consider including key pioneering studies such as: García BA, Canale DM, Blanco A. Genetic structure of four species of Triatoma (Hemiptera: Reduviidae) from Argentina. J Med Entomol. 1995; 32(2): 134-7. doi: 10.1093/jmedent/32.2.134, and García BA, Powell JR. Phylogeny of species of Triatoma (Hemiptera: Reduviidae) based on mitochondrial DNA sequences. J Med Entomol. 1998; 35(3): 232-8. doi: 10.1093/jmedent/35.3.232.

Our response: As the reviewer noted, these two studies are pioneers in the analysis of enzyme variability and mitochondrial DNA sequence data. Both papers have been incorporated into the Introduction section. They are cited in the first paragraph of the Introduction, which now reads as follows:

(...) These six species (T. rosai was referred as T. sordida Argentina) were grouped within the Sordida subcomplex through chromosomal markers (Pita et al. 2016), and later confirmed through analysis of nuclear and mitochondrial DNA sequences (Monteiro et al. 2018, Belintani et al. 2020). In 1998, García and Powell, using DNA sequences of mitochondrial genes 12S, 16S, and cytochrome c oxidase subunit I (coI), demonstrated that T. sordida from Argentina and Brazil form a well-supported clade, which appears as the sister taxon to the guasayana–rubrovaria–circummaculata clade. The initial inclusion of Triatoma guasayana (Wygodzinsky and Abalos, 1949) and Triatoma patagonica (Del Ponte, 1929) in this subcomplex, based on morphological similarities (Carcavallo et al. 2000, Schofield and Galvão 2009), has been discarded by numerous chromosomal, molecular and chemical evidences (Pita et al. 2016, Monteiro et al. 2018, Belintani et al. 2020, Moriconi et al. 2022). Enzyme variability in four Triatoma species from Argentina, including T. sordida, showed that this species exhibited a low level of heterozygosity compared to T. guasayana. The results of this study also indicated that T. sordida and T. guasayana form a closely related species pair (Garcia et al. 1995). Nevertheless, several authors based on mitochondrial and nuclear sequences, grouping T. guasayana within the Sordida subcomplex (Alevi et al. 2020; Kieran et al. 2021).

2. The first paragraph of the Discussion (line 387) would benefit from a brief summary of the study's aims and analytical approaches, to guide the reader and establish a clearer narrative transition from the results. As currently written, the section begins abruptly with conclusions, without restating the objectives or the rationale for the study.

Our response: We thank the reviewer for the suggestion. The text was changed (lines 424-431): "The Sordida subcomplex has long posed taxonomic challenges due to the morphological similarity among species, their overlapping distributions, and evidence of natural hybridization. To contribute to a clearer understanding of

this group, we analyzed a mitochondrial (cytb) gene fragment and chromosomal markers across 60 new populations from Argentina, Bolivia, Brazil, and Paraguay. Our results suggest the existence of at least six genetically distinct lineages — a diversity greater than currently recognized— and highlight regional patterns that were not evident from morphology alone." (...).

3. The incorrect molecular species identification section raises an important point by discussing potentially misidentified cytb sequences in GenBank. While a few accession numbers are mentioned in the main text (e.g., KC249290, KC249293, and KC249295), others are referenced more generally (e.g., "four T. guasayana sequences") without explicitly listing their accession numbers or providing supporting details for reclassification. Although these sequences are noted as highlighted in Supplementary Table S1, the authors might consider including a concise table — either in the main text or as supplementary material — summarizing: (i) the original GenBank accession, (ii) the initial species designation, (iii) the reassigned identity based on the current analyses, and (iv) brief supporting rationale. This addition could improve clarity and reproducibility for readers, though its inclusion is entirely at the authors' discretion.

Our response: We are in agreement and thank the reviewer for the suggestion. Within the manuscript we further explain the accession where we found issues. Now all the accessions are explained as the reviewer suggested: Which species is in GenBank and which species are we reporting that actually is. Together with the acc. number. The text was changed as follow: "According to our analysis, seven GenBank sequences were misidentified (Table S1, in red). Surprisingly, a T. sordida GenBank sequence (Hap. 1 —KC249290) was also included within the outgroup, presenting a high identity (98.71%) with T. infestans (Klug, 1834) (Acc. Number HQ333230). Two T. sordida GenBank sequences (KC249293 and KC249295) were identified as T. garciabesi and T. rosai, respectively. Finally, other four GenBank sequences erroneously identified as belonging to T. guasayana grouped into two clades within the Sordida subcomplex (T. sordida s.s. —Hap. 24: KC249250— and T. sordida La Paz —Hap. 50: MH054944; Hap. 52: KC249252 and KC249253) (Fig. 1A y Fig. S1)."

Reviewer #2

This MS represents an important and relevant study to reconstruct the Sordida subcomplex of species belonging to the Triatominae subfamily (Hemiptera, Reduviidae) being most of these taxa vectors of Chagas disease in South America. The initial hypothesis to test for the taxa integration of this subcomplex was correctly achieved. The abstract includes major findings of the present study. This research includes multidisciplinary approaches like molecular systematics, several parameters of population genetics based on Cytb mitochondrial gene, cytogenetics data and biogeography, including important number of samples collected among different location and countries where these taxa distribute in South America. Methodologically is a well conducted research, including robust phylogenetic analyses to support possible scenarios of differentiation for these species subcomplex. In this regard, my major concern refers to this topic. In my modest opinion, the authors could to include other additional analyses (i.e. Mistmach distribution, indirect estimate of gene flow, among others). Thus, the Discussion could be include possible scenarios of taxa differentiation, a historical biogeography hypothesis (patterns and processes) based in this robust analyses (or additional as the aforementiod ones) for the described and undescribed taxa included in this subcomplex. I think that to extend the objective of present paper including and discussing differentiation hypothesis of the Sordida subcomplex would improve it, as potential evolutionary model. Figures and tables, as well references, have the required information to interpret results, but the authors must to check the correct present format for them.

I have made more detailed comments (see in the attached MS revised pdf file and please display all corresponding notes), so I prefer do not repeat here.

Finally, I think that the authors must include the recommendations above mentioned, to obtain more conclusive scenarios of the differentiation for this species group to support biodiversity studies, in this interesting and important group.

Our response: We are deeply grateful about the reviewer's comments. Thank you for highlighting our work. In order to attend to the comments in the text, the new version has changes which were indicated in the pdf attached by the reviewer.

In addition, regarding the comment "In my modest opinion, the authors could to include other additional analyses (i.e. Mistmach distribution, indirect estimate of gene flow, among others)." These tests were carried out and included in the manuscript, methods, results and discussion sections.

Methods: The R core base version 3.6.1 (R core team 2013), vegan (https://CRAN.R-project.org/package=vegan), ape (Paradis et al. 2004) and pegas (Paradis 2010) packages were employed to calculate the within and intergroup pairwise Kimura two parameters (K2p) distance, Tajimas' D test (Tajima 1989) as well as nucleotide diversity ($\pi$), number of haplotypes (h), segregating sites (S) and haplotype diversity (H). In addition, the MMD function of the pegas package was used to draw a histogram of the frequencies of pairwise distances for each of the three largest lineages from the cytb dataset (T. sordida s.s., T. garciabesi and T. rosai), which represent the mismatch distribution plot. The adegenet and hierfstat R packages were utilized to calculate Fst statistics. Finally, we construct a minimum spanning tree (MST) haplotype network using PopART software (Leigh and Bryant 2015).

Results: lines 391-395: (...) The mismatch distribution plots over the most sampled lineages are showing two different patterns. For T. sordida s.s. and T.rosai a skewed histogram is seen where most individuals present similar haplotypes, with just a few divergent ones. On the other hand, T. garciabesi presents a bimodal plot (Fig. S4). lines 408-410: (...) In addition, Fst statistics revealed great differentiation between lineages, with values ranging from 0.78 to 0.96.

Hence, according to these new results, and also the comments of the reviewer ("the Discussion could be include possible scenarios of taxa differentiation, a historical biogeography hypothesis (patterns and processes) based in this robust analyses (or additional as the aforementiod ones) for the described and undescribed taxa included in this subcomplex") the discussion was changed to include new results. Also, the conclusions were grated modified as follow:

Lines 589-618: Our study not only provides a detailed molecular and chromosomal framework for understanding the Sordida subcomplex but also emphasizes the critical role of integrating multidisciplinary approaches to resolve taxonomic ambiguities. The accurate identification of species and clades within this complex is essential for understanding their evolutionary trajectories and their role in the transmission dynamics of Trypanosoma cruzi. The identification of six mitochondrial lineages —three of them currently undescribed— suggests a more complex evolutionary history than previously recognized. The distinct DNA sequence parameters (haplotype and nucleotide diversity, neutrality), mismatch distribution patterns, together with high Fst values and phylogenetic support, point to contrasting demographic scenarios and limited gene flow among lineages, consistent with ongoing or past speciation processes. These results also hint at possible historical biogeographic patterns of isolation and expansion across the Gran Chaco, Andean, and Atlantic forest regions. Misidentifications in public databases and the occurrence of natural hybrids further complicate species delimitation, reinforcing the need for robust integrative taxonomic frameworks. Although the limited resolution of mitochondrial data prevents a definitive evolutionary reconstruction, our findings allow us to propose a working hypothesis for the differentiation of the Sordida subcomplex. The expansion signals observed in T. sordida s.s. and T. rosai may reflect recent colonization events following ecological or anthropogenic changes, whereas the deep genetic structure and bimodal mismatch pattern in T. garciabesi suggest a different evolutionary history, potentially involving incipient speciation or historical fragmentation. Together, these patterns support the view that the Sordida subcomplex comprises lineages at different stages of divergence, shaped by heterogeneous demographic and biogeographic processes across South America. A full understanding of these dynamics will require broader sampling and the incorporation of nuclear genomic data. Finally, we propose the Sordida subcomplex as a promising model system to study speciation, hybridization, and biogeographic diversification in Chagas disease vectors.

## SECOND REVIEW ROUND

### REVIEWERS COMMENTS

### REVIEWER #1

The authors have effectively addressed all the reviewers' suggestions. In my opinion, the manuscript can be accepted for publication.

### REVIEWER #2

I consider the authors to have conducted a very thorough review of the MS, incorporating the suggestions made in the initial review. The additional analyses incorporated generate new hypotheses for future testing regarding the Sordida subcomplex, as the authors detect different levels of differentiation and speciation among the taxa currently included in this complex. This group, represents an interesting model for studying speciation in this group of insects so closely linked to Chagas disease. I find the review of the text in all its items, figures, and tables to be accurate, and in my opinion, has highlighted the importance of this topic.

