## [Reviewer Report · FIRST REVIEW ROUND - REVIEWERS COMMENTS]

## Reviewer #1

General comments

The manuscript “Reconstructing Sordida subcomplex (Hemiptera, Reduviidae, Triatominae) phylogeny across species distribution range”, authored by Gabriela Burgueño-Rodríguez and corresponding author Sebastián Pita, presents a comprehensive phylogenetic analysis of 95 male adult specimens from natural populations across 43 localities in Argentina. The study combines DNA sequence analyses, fluorescent in situ hybridization, and C-heterochromatin distribution (C-banding), while also revisiting previously published data and GenBank entries. A particularly valuable aspect of this work is the identification of potential misassignments in public sequence databases—an important contribution that may help prevent the replication of such errors in future research. The methodology is sound, and the results are clearly presented and thoroughly discussed. Overall, this is a well-executed and rigorous study, well within the scope of Memórias do Instituto Oswaldo Cruz. I recommend its publication with minor revisions. A few suggestions to improve the taxonomic and literature review sections are included as specific comments.

Specific comments

1. To provide a more complete historical context of the evolutionary systematics of the Sordida subcomplex, the authors should consider including key pioneering studies such as: García BA, Canale DM, Blanco A. Genetic structure of four species of Triatoma (Hemiptera: Reduviidae) from Argentina. J Med Entomol. 1995; 32(2): 134-7. doi: 10.1093/jmedent/32.2.134, and García BA, Powell JR. Phylogeny of species of Triatoma (Hemiptera: Reduviidae) based on mitochondrial DNA sequences. J Med Entomol. 1998; 35(3): 232-8. doi: 10.1093/jmedent/35.3.232.

2. The first paragraph of the Discussion (line 387) would benefit from a brief summary of the study’s aims and analytical approaches, to guide the reader and establish a clearer narrative transition from the results. As currently written, the section begins abruptly with conclusions, without restating the objectives or the rationale for the study.

3. The incorrect molecular species identification section raises an important point by discussing potentially misidentified cytb sequences in GenBank. While a few accession numbers are mentioned in the main text (e.g., KC249290, KC249293, and KC249295), others are referenced more generally (e.g., “four T. guasayana sequences”) without explicitly listing their accession numbers or providing supporting details for reclassification. Although these sequences are noted as highlighted in Supplementary Table S1, the authors might consider including a concise table — either in the main text or as supplementary material—summarizing: (i) the original GenBank accession, (ii) the initial species designation, (iii) the reassigned identity based on the current analyses, and (iv) brief supporting rationale. This addition could improve clarity and reproducibility for readers, though its inclusion is entirely at the authors’ discretion.

## Reviewer #2

This MS represents an important and relevant study to reconstruct the Sordida subcomplex of species belonging to the Triatominae subfamily (Hemiptera, Reduviidae) being most of these taxa vectors of Chagas disease in South America. The initial hypothesis to test for the taxa integration of this subcomplex was correctly achieved. The abstract includes major findings of the present study. This research includes multidisciplinary approaches like molecular systematics, several parameters of population genetics based on Cytb mitochondrial gene, cytogenetics data and biogeography, including important number of samples collected among different location and countries where these taxa distribute in South America. Methodologically is a well conducted research, including robust phylogenetic analyses to support possible scenarios of differentiation for these species subcomplex. In this regard, my major concern refers to this topic. In my modest opinion, the authors could to include other additional analyses (i.e. Mistmach distribution, indirect estimate of gene flow, among others). Thus, the Discussion could be include possible scenarios of taxa differentiation, a historical biogeography hypothesis (patterns and processes) based in this robust analyses (or additional as the aforementiod ones) for the described and undescribed taxa included in this subcomplex. I think that to extend the objective of present paper including and discussing differentiation hypothesis of the Sordida subcomplex would improve it, as potential evolutionary model. Figures and tables, as well references, have the required information to interpret results, but the authors must to check the correct present format for them.

I have made more detailed comments (see in the attached MS revised pdf file and please display all corresponding notes), so I prefer do not repeat here.

Finally, I think that the authors must include the recommendations above mentioned, to obtain more conclusive scenarios of the differentiation for this species group to support biodiversity studies, in this interesting and important group.

---

## [Author Response · AUTHORS RESPONSE TO REVIEWERS]

## Dr Adeilton Alves Brandão

Editor in chief

Dr Adeilton Brandão

Handling Editor

Memórias do Instituto Oswaldo Cruz

Thank you very much for the opportunity to resubmit our manuscript. We have revised the reviewers’ comments. We are very grateful for the constructive comments. We have taken all the comments seriously and have made as many suggested changes as possible.

We have tried to respond adequately to all the issues raised by both reviewers.

Thank you for your time and consideration. We look forward to hearing from you.

Sincerely,

Sebastián Pita, Ph.D.

Below we respond to each of the reviewer’s comments and the modifications carried out in the revised manuscript:

## Reviewer #1

General Comments

The manuscript “Reconstructing Sordida subcomplex (Hemiptera, Reduviidae, Triatominae) phylogeny across species distribution range”, authored by Gabriela Burgueño-Rodríguez and corresponding author Sebastián Pita, presents a comprehensive phylogenetic analysis of 95 male adult specimens from natural populations across 43 localities in Argentina. The study combines DNA sequence analyses, fluorescent in situ hybridization, and C-heterochromatin distribution (C-banding), while also revisiting previously published data and GenBank entries. A particularly valuable aspect of this work is the identification of potential misassignments in public sequence databases—an important contribution that may help prevent the replication of such errors in future research. The methodology is sound, and the results are clearly presented and thoroughly discussed. Overall, this is a well-executed and rigorous study, well within the scope of Memórias do Instituto Oswaldo Cruz. I recommend its publication with minor revisions. A few suggestions to improve the taxonomic and literature review sections are included as specific comments.

Specific comments

1. To provide a more complete historical context of the evolutionary systematics of the Sordida subcomplex, the authors should consider including key pioneering studies such as: García BA, Canale DM, Blanco A. Genetic structure of four species of Triatoma (Hemiptera: Reduviidae) from Argentina. J Med Entomol. 1995; 32(2): 134-7. doi: 10.1093/jmedent/32.2.134, and García BA, Powell JR. Phylogeny of species of Triatoma (Hemiptera: Reduviidae) based on mitochondrial DNA sequences. J Med Entomol. 1998; 35(3): 232-8. doi: 10.1093/jmedent/35.3.232.

Our response: As the reviewer noted, these two studies are pioneers in the analysis of enzyme variability and mitochondrial DNA sequence data. Both papers have been incorporated into the Introduction section. They are cited in the first paragraph of the Introduction, which now reads as follows:

(...) These six species (T. rosai was referred as T. sordida Argentina) were grouped within the Sordida subcomplex through chromosomal markers (Pita et al. 2016), and later confirmed through analysis of nuclear and mitochondrial DNA sequences (Monteiro et al. 2018, Belintani et al. 2020). In 1998, García and Powell, using DNA sequences of mitochondrial genes 12S, 16S, and cytochrome c oxidase subunit I (coI), demonstrated that T. sordida from Argentina and Brazil form a well-supported clade, which appears as the sister taxon to the guasayana–rubrovaria–circummaculata clade. The initial inclusion of Triatoma guasayana (Wygodzinsky and Abalos, 1949) and Triatoma patagonica (Del Ponte, 1929) in this subcomplex, based on morphological similarities (Carcavallo et al. 2000, Schofield and Galvão 2009), has been discarded by numerous chromosomal, molecular and chemical evidences (Pita et al. 2016, Monteiro et al. 2018, Belintani et al. 2020, Moriconi et al. 2022). Enzyme variability in four Triatoma species from Argentina, including T. sordida, showed that this species exhibited a low level of heterozygosity compared to T. guasayana. The results of this study also indicated that T. sordida and T. guasayana form a closely related species pair (Garcia et al. 1995). Nevertheless, several authors based on mitochondrial and nuclear sequences, grouping T. guasayana within the Sordida subcomplex (Alevi et al. 2020; Kieran et al. 2021).

2. The first paragraph of the Discussion (line 387) would benefit from a brief summary of the study’s aims and analytical approaches, to guide the reader and establish a clearer narrative transition from the results. As currently written, the section begins abruptly with conclusions, without restating the objectives or the rationale for the study.

Our response: We thank the reviewer for the suggestion. The text was changed (lines 424-431): “The Sordida subcomplex has long posed taxonomic challenges due to the morphological similarity among species, their overlapping distributions, and evidence of natural hybridization. To contribute to a clearer understanding of this group, we analyzed a mitochondrial (cytb) gene fragment and chromosomal markers across 60 new populations from Argentina, Bolivia, Brazil, and Paraguay. Our results suggest the existence of at least six genetically distinct lineages — a diversity greater than currently recognized— and highlight regional patterns that were not evident from morphology alone.” (...).

3. The incorrect molecular species identification section raises an important point by discussing potentially misidentified cytb sequences in GenBank. While a few accession numbers are mentioned in the main text (e.g., KC249290, KC249293, and KC249295), others are referenced more generally (e.g., “four T. guasayana sequences”) without explicitly listing their accession numbers or providing supporting details for reclassification. Although these sequences are noted as highlighted in Supplementary Table S1, the authors might consider including a concise table — either in the main text or as supplementary material — summarizing: (i) the original GenBank accession, (ii) the initial species designation, (iii) the reassigned identity based on the current analyses, and (iv) brief supporting rationale. This addition could improve clarity and reproducibility for readers, though its inclusion is entirely at the authors’ discretion.

Our response: We are in agreement and thank the reviewer for the suggestion. Within the manuscript we further explain the accession where we found issues. Now all the accessions are explained as the reviewer suggested: Which species is in GenBank and which species are we reporting that actually is. Together with the acc. number. The text was changed as follow: “According to our analysis, seven GenBank sequences were misidentified (Table S1, in red). Surprisingly, a T. sordida GenBank sequence (Hap. 1 —KC249290) was also included within the outgroup, presenting a high identity (98.71%) with T. infestans (Klug, 1834) (Acc. Number HQ333230). Two T. sordida GenBank sequences (KC249293 and KC249295) were identified as T. garciabesi and T. rosai, respectively. Finally, other four GenBank sequences erroneously identified as belonging to T. guasayana grouped into two clades within the Sordida subcomplex (T. sordida s.s. —Hap. 24: KC249250— and T. sordida La Paz —Hap. 50: MH054944; Hap. 52: KC249252 and KC249253) (Fig. 1A y Fig. S1).”

## Reviewer #2

This MS represents an important and relevant study to reconstruct the Sordida subcomplex of species belonging to the Triatominae subfamily (Hemiptera, Reduviidae) being most of these taxa vectors of Chagas disease in South America. The initial hypothesis to test for the taxa integration of this subcomplex was correctly achieved. The abstract includes major findings of the present study. This research includes multidisciplinary approaches like molecular systematics, several parameters of population genetics based on Cytb mitochondrial gene, cytogenetics data and biogeography, including important number of samples collected among different location and countries where these taxa distribute in South America. Methodologically is a well conducted research, including robust phylogenetic analyses to support possible scenarios of differentiation for these species subcomplex. In this regard, my major concern refers to this topic. In my modest opinion, the authors could to include other additional analyses (i.e. Mistmach distribution, indirect estimate of gene flow, among others). Thus, the Discussion could be include possible scenarios of taxa differentiation, a historical biogeography hypothesis (patterns and processes) based in this robust analyses (or additional as the aforementiod ones) for the described and undescribed taxa included in this subcomplex. I think that to extend the objective of present paper including and discussing differentiation hypothesis of the Sordida subcomplex would improve it, as potential evolutionary model. Figures and tables, as well references, have the required information to interpret results, but the authors must to check the correct present format for them.

I have made more detailed comments (see in the attached MS revised pdf file and please display all corresponding notes), so I prefer do not repeat here.

Finally, I think that the authors must include the recommendations above mentioned, to obtain more conclusive scenarios of the differentiation for this species group to support biodiversity studies, in this interesting and important group.

Our response: We are deeply grateful about the reviewer’s comments. Thank you for highlighting our work. In order to attend to the comments in the text, the new version has changes which were indicated in the pdf attached by the reviewer.

In addition, regarding the comment “In my modest opinion, the authors could to include other additional analyses (i.e. Mistmach distribution, indirect estimate of gene flow, among others).” These tests were carried out and included in the manuscript, methods, results and discussion sections.

Methods: The R core base version 3.6.1 (R core team 2013), vegan (https://CRAN.R-project.org/package=vegan), ape (Paradis et al. 2004) and pegas (Paradis 2010) packages were employed to calculate the within and intergroup pairwise Kimura two parameters (K2p) distance, Tajimas’ D test (Tajima 1989) as well as nucleotide diversity (π), number of haplotypes (h), segregating sites (S) and haplotype diversity (H). In addition, the MMD function of the pegas package was used to draw a histogram of the frequencies of pairwise distances for each of the three largest lineages from the cytb dataset (T. sordida s.s., T. garciabesi and T. rosai), which represent the mismatch distribution plot. The adegenet and hierfstat R packages were utilized to calculate Fst statistics. Finally, we construct a minimum spanning tree (MST) haplotype network using PopART software (Leigh and Bryant 2015).

Results: lines 391-395: (...) The mismatch distribution plots over the most sampled lineages are showing two different patterns. For T. sordida s.s. and T.rosai a skewed histogram is seen where most individuals present similar haplotypes, with just a few divergent ones. On the other hand, T. garciabesi presents a bimodal plot (Fig. S4). lines 408-410: (...) In addition, Fst statistics revealed great differentiation between lineages, with values ranging from 0.78 to 0.96.

Hence, according to these new results, and also the comments of the reviewer (“the Discussion could be include possible scenarios of taxa differentiation, a historical biogeography hypothesis (patterns and processes) based in this robust analyses (or additional as the aforementiod ones) for the described and undescribed taxa included in this subcomplex”) the discussion was changed to include new results. Also, the conclusions were grated modified as follow:

Lines 589-618: Our study not only provides a detailed molecular and chromosomal framework for understanding the Sordida subcomplex but also emphasizes the critical role of integrating multidisciplinary approaches to resolve taxonomic ambiguities. The accurate identification of species and clades within this complex is essential for understanding their evolutionary trajectories and their role in the transmission dynamics of Trypanosoma cruzi. The identification of six mitochondrial lineages —three of them currently undescribed— suggests a more complex evolutionary history than previously recognized. The distinct DNA sequence parameters (haplotype and nucleotide diversity, neutrality), mismatch distribution patterns, together with high Fst values and phylogenetic support, point to contrasting demographic scenarios and limited gene flow among lineages, consistent with ongoing or past speciation processes. These results also hint at possible historical biogeographic patterns of isolation and expansion across the Gran Chaco, Andean, and Atlantic forest regions. Misidentifications in public databases and the occurrence of natural hybrids further complicate species delimitation, reinforcing the need for robust integrative taxonomic frameworks. Although the limited resolution of mitochondrial data prevents a definitive evolutionary reconstruction, our findings allow us to propose a working hypothesis for the differentiation of the Sordida subcomplex. The expansion signals observed in T. sordida s.s. and T. rosai may reflect recent colonization events following ecological or anthropogenic changes, whereas the deep genetic structure and bimodal mismatch pattern in T. garciabesi suggest a different evolutionary history, potentially involving incipient speciation or historical fragmentation. Together, these patterns support the view that the Sordida subcomplex comprises lineages at different stages of divergence, shaped by heterogeneous demographic and biogeographic processes across South America. A full understanding of these dynamics will require broader sampling and the incorporation of nuclear genomic data. Finally, we propose the Sordida subcomplex as a promising model system to study speciation, hybridization, and biogeographic diversification in Chagas disease vectors.

---

## [Reviewer Report · REVIEWERS COMMENTS]

## Reviewer #1

The authors have effectively addressed all the reviewers’ suggestions. In my opinion, the manuscript can be accepted for publication.

## Reviewer #2

I consider the authors to have conducted a very thorough review of the MS, incorporating the suggestions made in the initial review. The additional analyses incorporated generate new hypotheses for future testing regarding the Sordida subcomplex, as the authors detect different levels of differentiation and speciation among the taxa currently included in this complex. This group, represents an interesting model for studying speciation in this group of insects so closely linked to Chagas disease. I find the review of the text in all its items, figures, and tables to be accurate, and in my opinion, has highlighted the importance of this topic.